Positive selection on human gamete-recognition genes

Hart Michael W. mwhart@sfu.ca 1
Stover Daryn A. 2
Guerra Vanessa 1
Mozaffari Sahar V. 3
Ober Carole 3
Mugal Carina F. 4
Kaj Ingemar 5
1 Department of Biological Sciences, Simon Fraser University , Burnaby , British Columbia , Canada
2 School of Mathematical and Natural Sciences, Arizona State University Colleges at Lake Havasu City , Lake Havasu City , AZ , USA
3 Department of Human Genetics, University of Chicago , Chicago , IL , USA
4 Department of Ecology and Genetics, Uppsala University , Uppsala , Sweden
5 Department of Mathematics, Uppsala University , Uppsala , Sweden
Berghout Joanne
Electronic publication date: 2018 Jan 11
Publication date: 2018
Volume: 6
Electronic Location ID: e4259
Received 2017 Sep 19; Accepted 2017 Dec 21
Copyright: ©2018 Hart et al.
Copyright year: 2018
Copyright holder: Hart et al.
License: This is an open access article distributed under the terms of the Creative Commons Attribution License, which permits unrestricted use, distribution, reproduction and adaptation in any medium and for any purpose provided that it is properly attributed. For attribution, the original author(s), title, publication source (PeerJ) and either DOI or URL of the article must be cited.
License URL: https://creativecommons.org/licenses/by/4.0/

Keywords: Fertilization, Zona pellucida, Linkage disequilibrium, Epistasis, Balancing selection

Funding: Natural Sciences and Engineering Research Council of Canada This work was supported by Discovery Grants from the Natural Sciences and Engineering Research Council of Canada to M Hart (No. 05404) and B Crespi (No. 06505). The funders had no role in study design, data collection and analysis, decision to publish, or preparation of the manuscript.

==============================
Coevolution of genes that encode interacting proteins expressed on the surfaces of sperm and eggs can lead to variation in reproductive compatibility between mates and reproductive isolation between members of different species. Previous studies in mice and other mammals have focused in particular on evidence for positive or diversifying selection that shapes the evolution of genes that encode sperm-binding proteins expressed in the egg coat or zona pellucida (ZP). By fitting phylogenetic models of codon evolution to data from the 1000 Genomes Project, we identified candidate sites evolving under diversifying selection in the human genes ZP3 and ZP2. We also identified one candidate site under positive selection in C4BPA, which encodes a repetitive protein similar to the mouse protein ZP3R that is expressed in the sperm head and binds to the ZP at fertilization. Results from several additional analyses that applied population genetic models to the same data were consistent with the hypothesis of selection on those candidate sites leading to coevolution of sperm- and egg-expressed genes. By contrast, we found no candidate sites under selection in a fourth gene (ZP1) that encodes an egg coat structural protein not directly involved in sperm binding. Finally, we found that two of the candidate sites (in C4BPA and ZP2) were correlated with variation in family size and birth rate among Hutterite couples, and those two candidate sites were also in linkage disequilibrium in the same Hutterite study population. All of these lines of evidence are consistent with predictions from a previously proposed hypothesis of balancing selection on epistatic interactions between C4BPA and ZP3 at fertilization that lead to the evolution of co-adapted allele pairs. Such patterns also suggest specific molecular traits that may be associated with both natural reproductive variation and clinical infertility.

Introduction

Proteins expressed on the surfaces of gametes mediate several types of cellular interactions at the time of fertilization, including chemoattraction of swimming sperm towards the egg, sperm binding to the egg extracellular coat, lysis or dissolution of the egg coat, and fusion of the gamete plasma membranes. Because those interactions may directly affect reproductive success, the genes encoding such proteins are sensitive to selection and often show evidence of diversifying or positive selection leading to high rates of amino acid substitution (Swanson et al., 2001; Swanson & Vacquier, 2002; Hamm et al., 2007; Clark et al., 2009; Vicens & Roldan, 2014). Different adaptive solutions to selection acting on variation in gamete attraction, binding, or fusion can lead to significant population differentiation within species (e.g., Hart et al., 2014), and are implicated in the origin or maintenance of reproductive isolation between species (Coyne & Orr, 2004; Palumbi, 2009; Lessios, 2011; Gavrilets, 2014). Understanding the nature of selection on mammalian genes that encode gamete-binding proteins may help to characterize the causes of natural variation in reproductive success, the potential for development of contraceptive vaccines (Lou et al., 1995; Delves, Lund & Roitt, 2002; Meeusen et al., 2007), or the targets of clinical treatment for infertility (Männikkö et al., 2005; Huang et al., 2014).

The human genome includes three functional members of the ZP domain gene family (ZP1, ZP2, and ZP3) that are shared with the mouse genome and encode the peptide portion of three glycoproteins that form the thick, fibrous extracellular coat or zona pellucida surrounding the mammalian egg (Hirohashi et al., 2008; Burnett et al., 2011; Avella, Xiong & Dean, 2013). The mouse genes that encode proteins known to interact with the egg coat (see Turner & Hoekstra, 2008a) include Zp3r (also called sp56; Buffone et al., 2008; Wassarman, 2009), which is expressed in sperm, localized to the acrosomal vesicle in the sperm head, and binds to the mouse egg coat (Buffone et al., 2008; Wassarman, 2009). The mouse Zp3r gene is homologous and syntenic with the human gene encoding the complement component four binding protein alpha (C4BPA; OMIM 120830, 1q32.2). In experimental studies, mouse proteins in the sperm head (including ZP3R) interact with a sperm-binding domain in the C-terminal region of ZP3 (Chen, Litscher & Wassarman, 1998; Williams et al., 2006) and another in the N-terminal region of ZP2 (Baibakov et al., 2012; Avella, Xiong & Dean, 2013), but the role and relative significance of ZP3 versus ZP2 in the specificity of mouse sperm binding is controversial (Litscher, Williams & Wassarman, 2009; Baibakov et al., 2012; Muro et al., 2012; Avella, Xiong & Dean, 2013).

Comparative evolutionary studies revealed that Zp3 has evolved under diversifying selection among species in some mammalian clades, and that the signal of positive selection in mouse Zp3 is strongest in the exon that encodes the C-terminal sperm-binding domain (Swanson et al., 2001; Turner & Hoekstra, 2006; Turner & Hoekstra, 2008b). Unfortunately, there is little complementary information about the evolution of Zp3r/sp56. In the genomes of rodents, Zp3r occurs with C4bpa in a cluster of genes called the regulator of complement activator (RCA) that encode many of the proteins involved in innate immunity (Mayilyan, 2012). Evolutionary relationships among genes related to Zp3r and C4bpa are complicated by gene duplications, functional divergence, or loss of function leading to pseudogenes (e.g., Rodríguez de Córdoba et al., 1994; De Villena & Rodríguez de Córdoba, 1995; Hillarp et al., 1997; Krushkal, Bat & Gigli, 2000). Comparative analyses show strong evidence of diversifying selection acting on many sites in C4BPA among humans and other great apes (Cagliani et al., 2016), but the evolution of Zp3r/sp56 among mice and other rodents has not been studied (see Morgan et al., 2010; Morgan et al., 2017). In general, it has been difficult to determine which sperm-expressed genes mediate the specificity or compatibility of sperm binding to the mammalian egg coat (Turner & Hoekstra, 2008a). The role of particular sperm-expressed genes or their epistatic interactions with ZP genes is controversial in mouse studies, which indicate that ZP3R binds to the egg coat but is not required for male fertility (Wassarman, 2009; Muro et al., 2012; Claw, George & Swanson, 2014; Wilburn & Swanson, 2016; Wang et al., 2017). The role of other genes in the RCA cluster related to Zp3r (including C4BPA) in fertilization has not been studied directly in humans or other primates.

Important evidence for selection acting on epistatic interactions between genes has come from measures of the strength of gametic disequilibrium, or associations between polymorphisms at physically unlinked loci that encode networks of interacting gene products (Sinervo & Svensson, 2002; Single et al., 2007; Qian, Zhou & Tang, 2015). Because such associations are expected to break down at meiosis, their persistence within populations can be ascribed to ongoing selection that favours some functionally advantageous combinations of allele pairs. For genes involved in gamete recognition, such allele pairs may be maintained by a fitness advantage at fertilization (Palumbi, 1999; Clark et al., 2009; Buzbas, Joyce & Rosenberg, 2011). Rohlfs, Swanson & Weir (2010) analyzed nonrandom associations among single nucleotide polymorphisms (SNPs) in 1,504 individuals from the 1958 British Birth Cohort study (Wellcome Trust Case Control Consortium, 2007). That analysis used two measures of gametic disequilibrium (or linkage disequilibrium, LD) between genotypes at SNPs in the genomic region near human C4BPA and near ZP3 (OMIM 182889, 7q11.23). Permutation tests and simulations of selection acting on tag SNPs suggested that some of the pairwise associations between two-locus genotypes were significantly stronger than expected, and might be accounted for by selection favouring particular combinations of C4BPA and ZP3 genotypes at fertilization. Rohlfs, Swanson & Weir (2010) proposed an evolutionary scenario in which sexual conflict between mates over the optimal rate of sperm-egg binding (fast for males, slow for females) leads to cyclical episodes of coevolution between male and female molecular adaptations. In this scenario, selection on male adaptations for enhancing fertilization rates (at C4BPA) is followed by selection on female countermeasures for reducing rates of sperm-egg contact and mitigating the lethal effects of polyspermy on offspring (at ZP3); both parts of this cycle are driven by negative frequency-dependent selection (a form of balancing selection); and the process results in the accumulation of coadapted pairs of C4BPA and ZP3 alleles in LD.

The scenario and mechanism for human C4BPA-ZP3 coevolution proposed by Rohlfs, Swanson & Weir (2010) can account for polymorphism in both genes and LD between them. That scenario is consistent with other examples of temporal or cyclical variation in selection on genes encoding gamete-binding proteins (Levitan, 2012; Wilburn & Swanson, 2016), the evidence for widespread balancing selection across the human genome (Andrés et al., 2009), the proposed role of such balancing selection in the maintenance of Zp3 polymorphisms in mouse populations in nature (Turner & Hoekstra, 2008b), and the evidence for ZP3-ZP3R functional interactions at the protein and cellular level during experimental fertilizations of mouse eggs in the lab (Litscher, Williams & Wassarman, 2009). This scenario is also consistent with the general predictions from analogous models of balancing selection acting on cooperating or cheating genotypes in systems of social conflict (similar to systems of sexual conflict), in which negative frequency dependence leads to an evolutionary arms race between adaptations for cooperation or for cheating, maintains polymorphism for both traits, and creates LD between them (Brockhurst et al., 2014; Ostrowski et al., 2015).

The specific targets of selection on functional features of sperm-egg binding that could lead to coevolution of C4BPA and ZP3 are expected to be nonsynonymous nucleotide polymorphisms that affect amino acid sequence, protein structure, and the functional interaction between proteins at fertilization. However, few of the tag SNPs analyzed by Rohlfs, Swanson & Weir (2010) were in exons: none of the seven tag SNPs in ZP3 were in coding sequences; two of nine tag SNPs in C4BPA were in coding sequences, but only one of these (rs4844573) was a nonsynonymous SNP encoding a C4BPA amino acid difference. As a result, most of these pairwise associations between SNPs may be indirectly implicated in C4BPA-ZP3 coevolution, but are probably not the direct targets of selection acting on functional variation between coding sequences expressed on the surfaces of gametes.

Here we use data and methods that complement and expand on those of Rohlfs, Swanson & Weir (2010) to revisit the evidence for diversifying selection (and in particular evidence for the effects of balancing selection) on and coevolution among human genes involved in sperm binding to the ZP. Using previously published genome sequences, we focused on coding sequence variation in C4BPA and ZP3, and in the ZP2 gene (OMIM 182888, 16p12.2) that is also implicated in sperm-egg binding (Baibakov et al., 2012; Avella, Xiong & Dean, 2013). As a negative control, we also analyzed ZP1 (OMIM 615774, 11q12.2), which encodes a structural protein that interacts with ZP3 and ZP2 to form the fibrous matrix of the zona pellucida but is not known to interact directly with sperm. We found one candidate site each under episodic diversifying selection in C4BPA, ZP3, and ZP2 (but not ZP1) in human populations. Other patterns of variation at those three candidate sites were consistent with the hypothesis of balancing selection favouring polymorphisms in those genes. We also found evidence of significant covariation between two of the candidate sites and reproductive success in a previously well-studied human founder population. Altogether the results were consistent with the hypothesis of some form of balancing selection leading to coevolution among these gamete-recognition genes (as proposed by Rohlfs, Swanson & Weir, 2010, and indicate targets of ongoing selection that affect human fertility.

Methods and Results

Overview

Our motivation for this study came from previous analyses of sperm- and egg-expressed genes with epistatic interactions at fertilization in sea stars, in which we found evidence of selection for amino acid differences between alleles from diverging conspecific populations (Sunday & Hart, 2013) and from closely-related species (Popovic et al., 2014; Patiño et al., 2016), and strong effects of those amino acid differences on sperm-egg compatibility and fertilization rates in laboratory experiments (Hart et al., 2014). Our goals in this study were similar: to identify specific sites under selection that could account for previously discovered LD between human C4BPA and ZP3 (Rohlfs, Swanson & Weir, 2010), extend those analyses to include human ZP2, and identify possible phenotypic effects of those polymorphisms on reproductive success. For those reasons, we used a work flow similar to our previous studies of sea star genes that mediate sperm-egg binding and reproductive success.

We first used a phylogenetic model of codon evolution designed to identify individual codons that have experienced episodic or diversifying selection by the estimation of relative rates of nonsynonymous (dN) and synonymous nucleotide changes (dS) mapped onto a phylogeny of relationships among sampled haplotypes. The effects of selection are estimated as the ratio of those relative rates (ω = dN∕dS): selection favouring the accumulation of amino acid differences at rates higher than expected from neutral processes (mutation and genetic drift) alone is inferred for codons with ω > 1. We carried out this analysis first because it provides a powerful method to identify specific candidate sites under selection, although such codon models were originally developed for comparative analyses of coding sequence divergence between reproductively isolated species (Kryazhimskiy & Plotkin, 2008; Mugal, Wolf & Kaj, 2014). These sensitive models of codon evolution can identify small numbers of candidate sites under diversifying selection, and distinguish them from a background of overall purifying selection (ω < 1) or neutral evolution (ω∼1) at other sites (Murrell et al., 2012). In these models, parallel nonsynonymous changes at several times in the evolutionary history of the gene or in several lineages of the gene tree make up much of the signal of high ω values associated with specific codons under episodic or diversifying positive selection. When applied to conspecific populations, high ω values might indicate ongoing diversifying (or balancing) selection. Important caveats about that application and interpretation are summarized below.

Second, we used population genetic methods to explore variation at candidate sites that might be consistent with the scenario of Rohlfs, Swanson & Weir (2010) that these genes evolve under selection for sperm-egg compatibility mediated by sexual conflicts of interest. We used several population genetic tests for selection, and we examined patterns of haplotype frequency variation for each gene. These population genetic analyses can be used to corroborate our results from models of codon evolution, and may also help to account for possible false positives that are known to be associated with some models of codon evolution when those models are fitted to data from conspecific populations (Kryazhimskiy & Plotkin, 2008; Mugal, Wolf & Kaj, 2014).

Third, we searched for covariation between candidate sites under selection and their effect on gamete compatibility. Because sperm-egg binding and fertilization rates are not as readily observed in humans as in sea stars or other organisms with external fertilization, we used proxies for gamete binding and fertilization success based on family size and birth rate data from a previously well-studied human founder population (Kosova et al., 2012).

At several steps in the study our choice of the next analysis to carry out depended on the observations or analytical results from a previous step in the study. This type of data-dependent analysis has been called a “garden of forking paths” (Gelman & Loken, 2014) in which the choices of analyses by the researcher (sometimes called “researcher degrees of freedom”; Simmons, Nelson & Simonsohn, 2011) are not specified in advance of the study. Researcher degrees of freedom can be used to preferentially report only positive results with significant P-values, and that practice may inflate the apparent strength of results and lead to a high expected rate of false positive discoveries (Forstmeier, Wagenmakers & Parker, 2017). For those reasons, in the sections below we present both the analytical method used and the results of that analysis together for each step in order to show and acknowledge the data-dependent nature of the work flow. Our discoveries should be interpreted as the basis for follow-up studies and analysis, rather than as conclusive tests of the balancing selection and coevolution scenario proposed by Rohlfs, Swanson & Weir (2010).

Phylogenetic models of codon evolution indicate episodic diversifying selection at three candidate sites

We obtained 2,184 full-length coding sequences for C4BPA, ZP3, ZP2, and ZP1 from 1,092 individual humans in the Phase 1 release (20110521) of the 1000 Genomes Project database (1000 Genomes Project Consortium, 2012), which includes samples of 14 populations from four continental groups including Asia (Japan, northern and southern China), Europe (Finland, Britain, Spain, Italy, and European Americans from Utah), the Americas (Puerto Rico, Mexico, Colombia), and Africa (Kenya, Nigeria, and African Americans in the southwestern USA). Population sample sizes in Phase 1 were 14–100 individuals (28–200 gene copies). We used the Phase 1 release because these data consist of phased haplotypes suitable for analysis using phylogenetic or coalescent models of codon evolution that employ gene trees. We analyzed the longer of two transcriptional variants for ZP3. Each alignment was visualized in the Se-Al aligner (Rambaut, 2002), trimmed to coding sequence only, and exported for downstream analysis.

We used blastn searches of the GenBank nonredundant nucleotide database, and the SMART tool (Letunic, Doerks & Bork, 2015) for characterizing protein sequence domains, to confirm the identity of each gene downloaded from the 1000 Genomes database; we used the UCSC genome browser (Kent et al., 2002) to confirm the synteny of each gene with the corresponding mouse reference genes. The C4BPA protein consists of eight complement control peptide domains (CCP, also known as sushi domains; reviewed by Ermert & Blom, 2016) each approximately 55–60 amino acids in length. The ZP genes each encode several distinctive protein domain types, including a short signal peptide, a ZP domain about 260 amino acids in length, a furin-type peptide cleavage sequence, and a propeptide following the cleavage motif that includes a transmembrane domain involved in cytoplasmic trafficking of the protein into the extracellular egg coat (reviewed by Monné & Jovine, 2011 and Wilburn & Swanson, 2016).

We used the branch-site model of codon evolution implemented in MEME (mixed-effects model of evolution; Murrell et al., 2012) to identify sites in each gene with high relative rates of nonsynonymous change that might evolve under diversifying selection. We used tools on the Datamonkey web interface (Delport et al., 2010) to screen each sequence alignment for recombination and to choose a model of nucleotide evolution (Kosakovsky Pond et al., 2006). We then estimated a phylogeny using the maximum-likelihood method (and the best-fit nucleotide model) in MEGA 5 (Tamura et al., 2011), and used that gene tree in a MEME analysis. The MEME method is similar to the branch-sites models in the PAML software package (Yang, 2007), but has several advantages including the ability to model variation among codons in both the synonymous and nonsynonymous rates of change, and greater sensitivity to identify targets of selection (ω > 1) at a small number of functionally important codons among only some lineages in a gene tree.

The MEME analyses identified 1–3 codons in each of the three genes known to be involved in sperm-egg binding (C4BPA, ZP3, ZP2) that showed evidence of episodic diversifying selection (positive selection on those individual codons among some lineages in the gene tree; Table 1). For each of those codons, a likelihood ratio test indicated a significantly better model fit (P < 0.05) when the nonsynonymous rate β at those sites was allowed to exceed the synonymous rate α in comparison to a null model with the constraint β = α.

Table 1 Site-specific parameter values estimated in a model of codon evolution fitted to alignments of human gamete-recognition genes.

Site-specific parameter values for mixed-effects models of evolution (MEME; Murrell et al., 2012) are shown for five codons in four analyses. Parameters include α, the synonymous rate of change; β−, the nonsynonymous rate at that site for lineages in the constrained class of codons with β ≤ α; and β+, the unconstrained nonsynonymous rate (for codons in the positively-selected lineages of the gene tree). The probability of positive selection at each site (P) is calculated from the likelihood ratio test of models with the constrained value of β+ = α versus the unconstrained β+; q is the estimated false-discovery rate. Site-specific model parameter values are not shown for ZP1 (na) because no candidate sites were assigned to the positively selected class with β+ > α at P < 0.10.

Codon	MEME model parameter value	
	α	β−	β+	P	q	
C4PBA 300, rs4844573	0	0	184	0.0079	1	
ZP3 31, rs2286428	0	0	73.1	0.048	1	
ZP3 315, rs2906999	5.2 × 10−6	5.2 × 10−6	186	0.012	1	
ZP3 371, rs200481427	0	0	197	0.021	1	
ZP2 36, rs2075520	4.1 × 10−9	4.1 × 10−9	148	0.027	1	
ZP1	na	na	na	all P > 0.10		

In C4BPA there was one candidate site under selection (C4BPA 300; rs4844573) in the N-terminal region of the fifth sushi domain (Table 1) that showed a nonsynonymous polymorphism (a second-position transition encoding a threonine-isoleucine polymorphism; Table 2). Notably, this was the only nonsynonymous site included among the tag SNPs analyzed by Rohlfs, Swanson & Weir (2010), and was also one of the sites found by Rohlfs, Swanson & Weir (2010) to be in LD with a polymorphism near ZP3.

There were three sites under selection in ZP3 (Table 1), including one near the signal sequence (ZP3 31; rs2286428) in a protein domain of unknown function that is not part of the modeled protein structure based on X-ray crystallography of chicken ZP3 (Han et al., 2010; Monné & Jovine, 2011), and a second site in the C-terminal propeptide region (ZP3 371; rs200481427) downstream of the peptide cleavage site and thus not part of the mature functional protein in the egg coat. Because these two sites are in protein domains not expected to be involved in sperm binding, we focused our subsequent analyses of ZP3 on a third candidate site (ZP3 315; rs2906999) that showed a nonsynonymous polymorphism (a first-position transition encoding a proline-serine polymorphism) in the known sperm-binding domain (Table 2). In the structural model of chicken ZP3 by Han et al. (2010), this candidate site is in a C-terminal region of secondary structure that is strongly constrained by three disulphide bridges among six highly conserved cysteine residues upstream (C6, C8) and downstream (C9 − C12) of human ZP3 315 (see Monné & Jovine, 2011).

Table 2 Genomic organization of human gamete-recognition genes including candidate sites under selection.

Genome coordinates and gene structure are those in the UCSC genome browser. Ancestral alleles were identified by comparison to reference genes for other great apes.

Gene	GRCh38/hg38 Coordinates	Number of exons	Number of codons	Candidate site under selection	Ancestral Allele	Derived Allele	
C4BPA	chr1:207,113,026- 207,144,717	11	597	rs4844573 chr1:207,131,555 C4BPA 300	ACT Thr	ATT Ile	
ZP3	chr7:76,424,965- 76,442,056	8	424	rs2906999 chr7:76,440,494 ZP3 315	CCG Pro	TCG Ser	
ZP2	chr16:21,197,480- 21,211,547	19	745	rs2075520 chr16:21,211,351 ZP2 36	GTG Val	GGG Gly	
ZP1	Chr11:60,867,562- 60,875,656	12	638	None			

There was one candidate site under selection in ZP2 in the N-terminal part of the predicted protein (ZP2 36; rs2075520) that showed a nonsynonymous polymorphism (a second-position transversion encoding a valine-glycine polymorphism; Table 2). Like the candidate site in ZP3 noted above, this single codon under selection in ZP2 was also in the known sperm-binding domain.

In contrast to those results for three genes that are expected to be involved in sperm-egg binding, we found no candidate sites under selection in MEME analyses of ZP1 variation. All likelihood ratio tests suggested that the nonsynonymous rate β was not significantly different from the synonymous rate at any individual codon (all P > 0.10). That result suggests that diversifying selection acting on ZP genes may be limited to those genes (ZP3, ZP2) known to be involved in sperm binding. For those reasons we focused the remainder of our analyses on the three candidate sites under selection in C4BPA, ZP3, and ZP2.

Population genetic analyses also indicate selection at the same candidate sites

Models of codon evolution like MEME were originally developed to analyze sequences from highly divergent lineages such as reproductively isolated species that have diverged on a long time scale and for which phylogenetic relationships can be characterized accurately (Goldman & Yang, 1994; Muse & Gaut, 1994). An important model assumption is that differences between those sampled lineages are fixed substitutions (possibly caused by selection) and not segregating polymorphisms (influenced by genetic drift). False-positive results can arise when the model parameters are estimated from alignments of closely-related conspecific haplotypes that have diverged on short time scales: if the data include segregating nonsynonymous polymorphisms that arose as recent mutations under weak purifying selection with insufficient time for selection to eliminate those mutations, then a signal of high ω may be falsely ascribed to selection favouring the accumulation of nonsynonymous differences at some codons or in some lineages. Similarly, models of codon evolution (like MEME) based on phylogenies may be misled by uncertainty about relationships among closely-related haplotypes sampled within a single species, and an important source of such uncertainty is recombination. These potential problems of false discovery in codon-model analyses of population data are well known (Stajich & Hahn, 2005; Hofer et al., 2009; Kryazhimskiy & Plotkin, 2008; Anisimova & Liberles, 2012; Mugal, Wolf & Kaj, 2014). The high q values in Table 2 (q = 1 for all five sites with high ω values identified by the MEME method) are also consistent with the expected high false-discovery rate, and suggest that our MEME results could include some false positives.

We used three approaches to test the reliability of our MEME results. First, we used an additional model of codon evolution called omegaMap v.0.5 (Wilson & McVean, 2006) that bridges the gap between interspecific analyses of selection based on phylogenies and intraspecific analyses of selection based on population samples. The omegaMap model is fitted to data by estimating mutation rate and recombination rate parameters as well as ω for different regions of a gene alignment using a sliding window method. Rather than mapping those rates onto a single phylogenetic tree, omegaMap uses a population genetic approximation of the coalescent within a Bayesian framework that accounts for uncertainty about genealogical relationships among haplotypes.

The omegaMap algorithm is computationally intensive and would not converge on a model fit using the full alignment for any of the three genes we analyzed. Instead we analyzed a reduced dataset for each gene consisting of the two haplotypes for one individual selected at random from each population in the 1000 Genomes Phase 1 release. All omegaMap results reported were based on this reduced dataset of 28 haplotypes total. We used inverse or improper inverse priors for model parameters, and allowed both the selection and recombination parameters to vary along the alignment. We carried out a series of preliminary analyses to find suitable prior ranges, parameter starting values, burn-in lengths, and block widths for the sliding-window analyses. We then ran the final MCMC model fit for 500,000 or 700,000 iterations, and sampled the model parameter values at intervals of 500 or 700 steps.

In comparison to MEME model parameter values (Table 2), omegaMap identified the same candidate sites under diversifying selection, but gave much lower (and more realistic) estimates of ω (2–4; Fig. 1). The posterior probabilities of positive selection (ω > 1) at the three candidate sites were high (P > 0.8; Fig. 1). Those results suggest that our MEME analyses correctly identified candidate sites under selection, but may have overestimated the magnitude of the response to selection.

Figure 1 Positive selection at individual codons in human gamete-recognition genes.

A single-population coalescent model of selection and recombination (omegaMap; Wilson & McVean, 2006) identified the same candidate sites under positive selection that were identified by a branch-site model of codon evolution (MEME; Table 1). Values of ω were estimated using the sliding window method for alignments of haplotypes from the 1000 Genomes Phase 1 data for three gamete-recognition genes, including (A) the sperm-expressed gene C4BPA, and the egg coat glycoprotein genes (B) ZP3 and (C) ZP2. Posterior probabilities of positive selection are shown for three candidate sites (see Table 1).

Second, we used the 1000 Genomes Selection Browser v.1.0 (http://hsb.upf.edu; Pybus et al., 2014) to carry out population genetic tests for selection. This tool implements a series of selection analyses applied to population samples from Caucasian Europeans in Utah (CEU), Han Chinese from Beijing (CHB), and Yorubans from Ibidan, Nigeria (YRI), that are representative of continental population groups in the 1000 Genomes database. Among the methods available using this tool, we chose two based on population allele frequency differences including pairwise FST between populations and global FST among all three populations (Weir & Cockerham, 1984); two based on linkage disequilibrium among sites within each gene including the integrated haplotype score within each population sample (iHS; Voight et al., 2006), and extended haplotype homozygosity in comparisons between pairs of populations (XP-EHH; Sabeti et al., 2007; see also Zhong et al., 2011); and three based on the allele frequency spectrum within each population including Tajima’s D (Tajima, 1989), the composite likelihood ratio test (CLR; Nielsen et al., 2005), and Fay & Wu’s H (Fay & Wu, 2000). Although FST values for a single locus may be interpreted as measures of population differentiation when compared to a neutral expectation (based on permutation), such values can also be interpreted as indices of selection when compared to other loci across the genome (sometimes called outlier tests; Foll & Gaggiotti, 2008; Holsinger & Weir, 2009). All seven tests are designed to detect significant departures from expectations based on the effects of neutral processes alone (mutation, genetic drift, recombination; reviewed by Vitti, Grossman & Sabeti, 2013). They are typically interpreted as tests of directional or purifying selection leading to differences among populations, but the tests may also be sensitive to the effects of diversifying or balancing selection when the results of such selection also resemble an incomplete selective sweep. We analyzed a 2 Mbp interval around each candidate site under selection in order to characterize the evidence for selection associated with all variable sites. The 1000 Genomes Selection Browser estimates the P-value associated with each test statistic for each candidate site under selection as its rank score relative to other sites in the 2 Mbp interval (the proportion of all other sites with a test statistic value as high as or higher than the candidate site). Because that database uses the GRCh37/hg19 reference genome, we centered each 2 Mbp interval around the coordinates for each of the candidate sites under selection in that reference, including chr1:207,304,900 (rs4844573, C4BPA), chr7:75069811-77069811 (rs2906999, ZP3), and chr16:21,222,672 (rs2075520, ZP2), instead of the coordinates shown in Table 2 for each of those sites (in the GRCh38/hg38 reference genome).

We found evidence for selection acting on those three candidate sites from two of the three types of tests. Four of the tests based on population differentiation (three of six pairwise FST values, and one of two global FST values) suggested significant population allele frequency differences at both the candidate site in C4BPA and the candidate site in ZP2 (Table 3). All three of the significant results for pairwise FST tests involved comparisons between the African sample (YRI) and one of the non-African samples (for ZP2) or both of the non-African samples (for C4BPA), and for C4BPA we found significant global differentiation as well. Overall evidence for selection was stronger for the candidate site in C4BPA (three of four significant test results) than for the candidate site in ZP2 (one of four).

Table 3 Population genetic tests of selection on candidate sites.

Results for three types of tests are shown based on (A) population differentiation, (B) linkage disequilibrium, or (C) the allele frequency spectrum. Each entry is the score or test statistic value (calculated using the 1000 Genomes Selection Browser v.1.0) for each candidate site under selection in a representative European (CEU), Asian (CHB), or African (YRI) population, or for each population pair (in pairwise FST and XP-EHH tests). Probability values are shown only for seven significant test results with 0.01 < P < 0.05 based on the rank of each score relative to all other scores for other sites in a 2 Mbp interval around the candidate site; all other test results were not significantly different from the null expectation (P > 0.05). Note that tests for the third candidate site (ZP3 315, rs2906999) could not be carried out using the 1000 Genomes Selection Browser v.1.0 because that site was not coded as variable in the 1000 Genomes Selection Browser database. Comparable tests for that missing site (using other methods or software) are given in the main text.

(A) Population differentiation	Pairwise FST	Global FST	
Gene, SNP	Population	CHB	YRI		
C4PBA 300, rs4844573	CEU	−0.005	0.323 (P = 0.030)	0.238 (P = 0.048)	
	CHB		0.342 (P = 0.036)	
ZP2 36, rs2075520	CEU	0.059	0.337 (P = 0.027)	0.187	
	CHB		0.142	
(B) Linkage disequilibrium	XP-EHH	iHS	
Gene, SNP	Population	CHB	YRI		
C4PBA 300, rs4844573	CEU	−0.564	1.935 (P = 0.045)	0.594	
	CHB		2.601 (P = 0.017)	0.067	
	YRI			2.325 (P = 0.037)	
ZP2 36, rs2075520	CEU	0.017	−0.884	0.446	
	CHB		−0.488	0.140	
	YRI			1.129	
(C) Allele frequency spectrum	Tajima’s D	CLR	Fay & Wu’s H	
Gene, SNP	Population				
C4PBA 300, rs4844573	CEU	0.716	0.496	−10.432	
	CHB	1.491	0.152	−4.853	
	YRI	0.383	0.340	3.263	
ZP2 36, rs2075520	CEU	1.084	0.621	4.754	
	CHB	1.897	0.000	−4.736	
	YRI	−0.063	1.301	−2.453	

We also found evidence of selection in some tests based on measures of LD across sites within the C4BPA and ZP2 genes. Like the tests based on population differentiation, evidence for selection was limited to variation within the African sample (iHS) and comparisons between the African and non-African samples (XP-EHH) (Table 3), and was stronger for the candidate site in C4BPA (three of six significant test results) than for the candidate site in ZP2 (no significant test results). This evidence for selection acting on C4BPA variation seems convincing because the two tests differ in their sensitivity to detect homozygosity and LD caused by selection acting in the recent (iHS) or the more distant past (XP-EHH) (Vitti, Grossman & Sabeti, 2013), but both tests indicated significantly greater extent of homozygosity around rs4844573 in C4BPA compared to expected extent of homozygosity under neutral processes alone.

By contrast, none of the individual tests based on properties of the allele frequency spectrum (Table 3) suggested significant departures from neutral evolution for the candidate sites in C4BPA or ZP2. Values for all three test statistics were small within each of the three population samples, and none were unusual (all P > 0.09) relative to test statistic values for other variable sites in the 2 Mbp interval surrounding both of the candidate sites.

Unfortunately, the candidate site under selection in ZP3 (rs2906999) was not coded as a variable site in the 1000 Genomes Selection Browser database, so we used other methods to implement some of the same tests for that missing candidate site. We focused on the two classes of tests (based on population differentiation, and based on LD) that did indicate some evidence of selection on the other two candidate sites in tests using the 1000 Genomes Selection Browser. We calculated pairwise FST between the CEU, CHB, and YRI populations (all FST < 0.040) and global FST among all three populations (FST = 0.021) for the candidate site in ZP3 using the method of Raymond & Rousset (1995). When compared to the ranked scores for FST values at other sites in the 2 Mbp interval spanning that candidate site (rs2906999), the global (P = 0.53) and all pairwise FST comparisons (all P > 0.38) were not significant. Those comparisons do not indicate selection on the candidate site in ZP3 leading to population allele frequency differences.

We also used the software package selscan (Szpiech & Hernandez, 2014) to calculate iHS and XP-EHH values for the candidate site in ZP3. Unlike the 1000 Genomes Selection Browser, selscan calculates an unstandardized test statistic value (for iHS, or XP-EHH) and then characterizes the significance of that value by binning together SNPs with similar allele frequencies (we used 20 bins), and calculating a normalized test statistic value (a z-score) relative to normally-distributed test statistic values for all other SNPs in the same bin with a mean of zero and standard deviation of one. Among those six tests (three iHS values for individual populations, three pairwise XP-EHH values), only one was significant: the iHS score for the CHB population sample (iHS = 2.423, z = 2.353, P = 0.018). Other iHS scores for the candidate site in ZP3 were lower (iHS < 2.05, P > 0.05), and all XP-EHH scores were low (<0.6) and not significant (P > 0.6). Like the FST tests, the analyses based on LD provide only limited evidence of selection on the candidate site in ZP3 leading to extensive homozygosity around that site.

Third, we used McDonald–Kreitman tests (McDonald & Kreitman, 1991) to assess the relative frequency of non-synonymous polymorphisms within human populations, and compare them to the relative frequency of non-synonymous fixed substitution differences between humans and chimpanzees. These tests are typically interpreted as evidence of positive selection favouring functional (nonsynymous) divergence between species when the tests reveal a high relative rate of interspecific nonsynonymous substitutions. However, the same test can also be used to identify evidence of balancing selection favouring the maintenance of functional (nonsynonymous) diversity within species in the form of a high relative rate of intraspecific nonsynonymous polymorphisms (e.g., Schoville, Flowers & Burton, 2012). The test is conservative because it compiles evidence across all sites into a single test statistic for each gene, so a significant result would be strong evidence for balancing selection, but nonsignificant results might be ascribed to weak sensitivity to detect the effects of selection on one candidate site in each gene. We used a single chimpanzee (Pan troglodytes) haplotype from GenBank for each gene (XM_514156.4, XM_016929619.1, XM_009453354.2); the chimpanzee and human coding sequences were the same length for each gene, so no additional alignment was required. We used the method of Egea, Casillas & Barbadilla (2008) to calculate the test statistic and associated P values. For all three genes, a large proportion (63–73%) of human polymorphisms was nonsynonymous (Table S1). However, in all three tests the proportion of nonsynonymous polymorphisms was not significantly different from the proportion of nonsynonymous fixed differences in the 2 × 2 contingency table (all P > 0.18), thus none of the tests gave evidence of balancing selection.

Unlike Rohlfs, Swanson & Weir (2010), we did not use measures of LD based on allelic or genotypic associations between pairs of candidate sites themselves in the 1000 Genomes Project data to assess the strength of selection acting on those three genes because such tests were expected to be weak in this case. Brown (1975) showed that, for simple allelic associations, the power to reject the null hypothesis of linkage equilibrium was low for population sample sizes like those in Phase 1 of the 1000 Genomes Project (all samples ≤100), unless the true underlying LD was very strong. Similarly, Rohlfs, Swanson & Weir (2010) showed that the power to detect genotypic associations was also low for small samples (<200) unless the value of the selection coefficient for genotypic associations was high (s > 0.4). We avoided the creation of larger sample sizes by pooling population samples from the 1000 Genomes Project data because such pooled samples may show LD among unlinked sites as an artifact of population differentiation rather than as a consequence of selection for allele or genotype pairs. For those reasons, we were not able to compare directly the patterns of LD among three candidate sites under selection in the (small) 1000 Genomes Project samples to the previously documented patterns of LD between two of those candidate sites in the (larger) 1958 British Birth Cohort sample (Rohlfs, Swanson & Weir, 2010).

All three candidate sites under selection are balanced polymorphisms

We used the parsimony network method in TCS (Clement, Posada & Crandall, 2000) to illustrate both the frequencies and phylogenetic relationships among all 2,184 gene copies for each gene. Each of those three networks included many closed circuits that indicated uncertainty about phylogenetic relationships among closely related gene copies. To illustrate the patterns more clearly, we pruned each of the closed circuits (and reduced the network for each gene down to one example genealogy) by favouring connections to more common haplotypes (and pruning edges leading to less common haplotypes).

Figure 2 Gene genealogies of human gamete-recognition genes.

Parsimony networks of relationships among haplotypes were similar for three genes including (A) the sperm-expressed gene C4BPA, and the egg coat glycoprotein genes (B) ZP3 and (C) ZP2. Each diagram shows one of many most parsimonious genealogies, in which ambiguous relationships were represented by closed circuits in the parsimony network estimated by the TCS method (Clement, Posada & Crandall, 2000). For illustration, each closed circuit in the network was resolved to a single genealogy by favouring connections to more common haplotypes and pruning connections to less common haplotypes. Each square symbol represents a unique haplotype (44–65 per gene); the size of each square is proportional to relative frequency (1–300 gene copies out of 2,184 total); large bold squares indicate common haplotypes that occurred >300 times. Lines show single nucleotide differences between haplotypes; red lines show nonsynonymous changes at one candidate site under selection in each gene (Table 2; Fig. 1). The ancestral amino acid state at each of those positively-selected codons (see Table 2) is indicated for the common haplotype in that genealogy, and the direction of each substitution at the positively-selected site is indicated for each red line, including two parallel changes to the derived state (from proline to serine at ZP3 codon 315; from threonine to isoleucine at C4BPA codon 300) and one reversal to the ancestral state (from glycine to valine at ZP2 codon 36). For the two common haplotypes that differed by the non-synonymous polymorphism at the candidate site under selection, counts of that haplotype are shown for the four continental population groups from Europe (EUR), Asia (ASN), the Americas (AMR), and Africa (AFR) in the 1000 Genomes Phase 1 release.

We found similar patterns of frequency variation and phylogenetic relationships among haplotypes across all three genes (Fig. 2). In both C4BPA and ZP3 (Figs. 2A and 2B), there were two most common haplotypes. The two C4BPA common haplotypes differed only by the nonsynonymous difference at the candidate site in the fifth sushi domain (C4BPA 300); they accounted for 36.7% and 35.4% of 2,184 gene copies, compared to the much lower frequencies of 45 other haplotypes. Similarly, the two ZP3 common haplotypes differed from each other only by the nonsynonymous difference at the candidate site in the ZP3 sperm-binding domain (ZP3 315), and accounted for 35.9% and 26.6% of all chromosomes, compared to lower frequencies of 63 other haplotypes.

We found a slightly more complex pattern of variation among ZP2 haplotypes (Fig. 2C), including four common haplotypes plus 40 less common haplotypes. A single nonsynonymous difference at the candidate site in the ZP2 sperm-binding domain (ZP2 36) distinguished the two pairs of common haplotypes from each other. The two common haplotypes distinguished by the candidate site under selection accounted for 14.0% and 26.1% of all chromosomes; each of those haplotypes differed from another common haplotype by a single synonymous change (at a different codon). The four common haplotypes together accounted for 92.3% of chromosomes. These patterns of diversity for each gene, characterized by a pair of common haplotypes that differ only by a single nonsynonymous difference at a candidate site under selection, are consistent with the predictions of the balancing selection hypothesis of Rohlfs, Swanson & Weir (2010).

In order to infer the direction of nucleotide changes at each candidate site in those phylogenetic networks, we used comparisons to other great ape reference genomes (chimpanzee, gorilla, orangutan) in the UCSC genome browser (Kent et al., 2002). We found that for each candidate site the chimpanzee, gorilla, and orangutan reference genomes as well as the mRNA sequences from bonobos all had the same single human allele (C4BPA 300 Thr; ZP3 315 Pro; ZP2 36 Val), which suggested that the alternate allele found in humans (C4BPA 300 Ile; ZP3 315 Ser; ZP2 36 Gly) is derived (Table 2). Mapping those ancestral and derived states onto the gene genealogies allowed us to infer the directions of changes (Fig. 2). In two cases (C4BPA, ZP2), the minor allele summed over all populations was also the ancestral allele; in the third case (ZP3), the ancestral and derived allele frequencies were nearly identical and the minor allele (by a small difference) was the derived allele (Table 4). All three genes showed multiple changes at the candidate site. On the gene trees shown in Fig. 2 these included two cases of parallel changes from the ancestral allele (C4BPA 300 Thr; ZP3 315 Pro) to the derived allele (C4BPA 300 Ile; ZP3 315 Ser), and one case of a secondary reversal from the derived allele (ZP2 36 Gly) back to the ancestral allele (ZP2 36 Val). Such parallelisms may account for the signal of selection found in our analyses using models of codon evolution (MEME, omegaMap) that are sensitive to the effects of multiple episodes of selection on the same codon (e.g., Turner & Hoekstra, 2006). Vahdati & Wagner (2016) argued that such parallelisms are a reliable signal of positive selection (but possibly not balancing selection) in genome scans of protein-coding sequences. The discovery of similar patterns of parallelism among C4BPA, ZP3, and ZP2 is also consistent with the hypothesis of selection acting on epistatic interactions among these three genes.

Table 4 Allele frequencies at candidate sites under selection in human gamete-recognition genes.

Entries on the left show frequencies for ancestral and derived alleles (ancestral alleles listed first) in four continental population groups in the 1000 Genomes Phase 1 release. Entries on the right show the minor allele frequency (MAF) summed over all populations, and the number of nonsynonymous SNPs in a 90.3 Mbp interval on the same chromosome that had a MAF greater than or equal to the candidate site (relative to N, the total number of nonsynonymous SNPS in parentheses). The significance of the MAF is characterized by P (the proportion of N with a MAF greater than or equal to the MAF for the candidate site).

Candidate site	Population group	MAF	SNPs (N)≥MAF	P	
		Africa	Americas	Asia	Europe				
C4BPA 300, rs4844573	ACT Thr	0.815	0.431	0.327	0.367	0.467	19 (1,408)	0.013	
	ATT Ile	0.185	0.569	0.673	0.633				
ZP3 315, rs2906999	CCG Pro	0.589	0.522	0.477	0.461				
	TCG Ser	0.411	0.478	0.523	0.538	0.495	1 (740)	0.0014	
ZP2 36, rs2075520	GTG Val	0.648	0.356	0.460	0.273	0.420	61 (1,445)	0.042	
	GGG Gly	0.352	0.644	0.540	0.727				

Balanced polymorphisms at candidate sites have unusually high minor allele frequencies

High frequencies for both ancestral and derived alleles, and minor allele frequency (MAF) differences between African and non-African populations, may arise by neutral demographic processes instead of (or in addition to) the effects of selection (Hofer et al., 2009; Coop et al., 2009). To explore that possibility, we carried out two tests.

First, we compared the observed MAFs at the three candidate sites under selection to the expected occurrence of sites with such high MAFs under a purely neutral model. This is a relatively stringent test given the expectation that nonsynonymous mutations in coding sequences are less likely than other mutations to evolve under neutrality. For each candidate site under selection we compared the observed range of MAFs in African versus non-African population groups (Table 4) to a neutral expectation based on a Poisson random field framework (Kaj & Mugal, 2016). We used the minimum MAFs for each gene from the four continental population groups (Table 4), and those MAFs (0.185, 0.411, 0.273) serve as a lower bound and thus make for a more conservative test.

We computed for each gene the expected number of sites at which the derived allele exists in both populations with an intermediate frequency in the range [x, 1 − x]. We let fx be defined by fx(y) = 1, if x < y < 1 − x, and fx(y) = 0 otherwise. Then fx ∈ F0 (see Kaj & Mugal, 2016), and the expected number of such sites has the form θHx, where (1) Hx= ∫01Py0x<ξt<1−x22ydy.

The function Py0x<ξt<1−x2 is known (in terms of hypergeometric sums) so we can find Hx numerically. We computed Hx for each gene with demographic conditions characteristic of the scenario for the late-Pleistocene migration of early modern humans out of Africa (population size N = 5, 000; population divergence time t = 0.5 coalescent units). Values of Hx were 0.8761, 0.0699, and 0.2149 for C4BPA, ZP3, and ZP2, respectively. We then used the numerical solution for Hx to calculate the probability p(x) = 1 − exp(−θHx) of observing at least one site that has intermediate allele frequencies in [x, 1 − x] under selectively neutral conditions for each of the three candidate sites (Table 4). For those calculations we used values of θ, the population mutation rate per gene, based on the 1,092 individuals of the 1000 Genomes phase 1 data (1.6333, 1.5727, and 1.9962 for C4BPA, ZP3, and ZP2, respectively). This approach leads to values for p(x) that are all relatively high (0.7609, 0.1042, 0.3488, respectively). Those high probabilities suggest that the observed MAFs for those three candidate sites in the African population group might have arisen by neutral processes alone under those demographic model conditions (the out-of-Africa migration of early modern humans).

Second, we compared MAFs at the three candidate sites under selection (summed over all populations) to MAFs at other nonsynonymous SNPs on the same three human chromosomes. In contrast to the analyses described above, this may be a less stringent (and more realistic) test given the expectation that most nonsynonymous mutations in coding sequences are likely to evolve under purifying selection. In order to standardize that comparison across candidate sites on three different chromosomes, we examined a window of 90.3 Mbp (the length of chromosome 16, the shortest of the three chromosomes studied here) spanning each of the candidate sites, and downloaded MAFs for all biallelic nonsynonymous SNPs in the 1000 Genomes Phase 1 release (3,593 SNPs total, including the three candidate sites). We characterized the significance of those balanced polymorphisms by estimating the proportion of other nonsynonymous SNPs on each chromosome that had MAFs greater than the MAF observed for the candidate site on that chromosome.

The high MAFs at each candidate site were unusual compared to all other nonsynonymous SNPs in a 90.3 Mbp window spanning each of those candidate sites. In all three cases, the frequency of the minor allele at the candidate site was higher than the MAFs at more than 95% of all other nonsynonymous SNPs on the same chromosome (Table 4). In the case of the ZP3 candidate site, no other nonsynonymous SNP in the same region of chromosome 7 showed such a high MAF. For each candidate site, the ancestral allele frequency tended to be higher in African samples compared to all non-African samples (Table 4), but both alleles at each candidate site were found in all populations. These unusual balanced polymorphisms are consistent with the predicted effects of episodic or diversifying selection that favours functionally divergent alleles at each candidate site, and maintains each allele at high frequency within populations. However, the differences between African and non-African population samples at all three candidate sites are also consistent with other hypotheses based on selection (such as an incomplete selective sweep in some or all populations).

The two tests together suggest that we cannot entirely reject neutrality as the cause of high observed MAFs at the three candidate sites (based on a stringent comparison to neutrality using calculations in the Poisson random field framework), but we find evidence for either relaxed selective constraints or diversifying selection (based on a more realistic comparison to expected variation under purifying selection at other nonsynonymous sites). Those results indicate that we should be cautious in interpreting the causes of allele frequency variation at those three candidate sites, but they do not contradict other evidence for diversifying selection acting on those three candidate sites.

Variation in fertility is associated with candidate sites under selection

To test for covariation between polymorphisms at candidate sites and measures of fertility we assessed genotype-phenotype associations in a founder population, the Hutterites (see Kosova, Abney & Ober, 2010; Kosova et al., 2010; Kosova et al., 2012), in which both reproductive data and genome-wide genotype data were available. As in previous analyses (e.g., Kosova et al., 2010), we focused on two measures of fertility: family size (number of births) in 361 families; and birth rate (calculated as [number of births − 1]/[sum of all interbirth intervals]) in 341 families with at least two children. We analyzed associations between these fertility measures and two of the three candidate sites (in C4BPA and ZP2); genotypes for the third candidate site (in ZP3) were not available for most individuals in the sample because that specific SNP (rs2906999) was not included in the SNP chip used to screen genetic polymorphisms in the original study (Kosova et al., 2012). We estimated genotype-phenotype associations using a linear mixed model as implemented in GEMMA (Zhou & Stephens, 2014). This model allowed us to account for the known pedigree relatedness among individuals in the sample. Following Kosova et al. (2012), we included wife’s birth year, her age at marriage, and the time from marriage to the most recent birth as covariates in the linear models because these covariates make substantial non-genetic contributions to fertility variation among Hutterite men and women. For each family we characterized the wife’s genotype at the ZP2 candidate site and the husband’s genotype at the C4BPA candidate site as having zero, one, or two derived alleles. We carried out a series of four parallel analyses in which we first asked whether variation in the wife’s ZP2 genotype or the husband’s C4BPA genotype was significantly associated with residual variation in family size or birth rate. We then repeated those analyses by including the spouse’s genotype at the other candidate site as an additional covariate, and by adding an interaction term for covariation between the two individual genotypes.

Although these post hoc analyses of functional variation were carried out after the identification of candidate sites under selection in C4BPA and ZP2, and the original genetic screens were not carried out with these genetic variants in mind, the analyses provide an important check on our inferences based on phylogenetic and population genetic methods. The models of codon evolution that we used to identify candidate sites under selection are only able to infer the effects of selection that acted in the past, and may have a high risk of false discoveries. One powerful source of evidence for ongoing (rather than past) balancing selection on such sites is their association with phenotypic variation related to fitness in present-day populations (Key et al., 2014). Such tests also can directly address the possibility that candidate sites under selection are false positives. Association studies in groups such as the Hutterites can be especially useful because fertility is generally high among Hutterite couples, and the Hutterites’ communal lifestyle and shared religious and cultural practices minimize variation in other non-genetic contributions to fertility (e.g., Kosova, Abney & Ober, 2010).

We found a significant association between the wife’s ZP2 36 genotype and variation in family size (P = 0.040) (Fig. 3A). Adding the husband’s C4BPA 300 genotype as a covariate, with or without an interaction term between the two genetic variables, did not improve the model fit and reduced the P value associated with the wife’s ZP2 36 genotype. There was no effect of the wife’s ZP2 36 genotype on birth rate, with or without an interaction term between the partners’ genotypes.

Figure 3 Fertility varies with genotype at two candidate sites under selection.

Hutterite families in which wives have more copies of the derived ZP2 36 Gly allele have fewer children (A), and families in which husbands have more copies of the derived C4BPA 300 Ile allele have higher birth rates (B). Horizontal bars show means for each genotype class. Plotted values are residuals from linear model analyses (GEMMA) that account for three other covariates (wife’s birth year, age at marriage, time from marriage to last birth); estimates of the statistical significance of those residual relationships also account for pedigree relatedness among families.

By contrast, there was a significant effect of the husband’s C4BPA 300 genotype on variation in birth rate (P = 0.020; Fig. 3B). Similar to the analyses described above, adding the wife’s ZP2 36 genotype as a covariate did not improve the model fit. The husband’s C4BPA 300 genotype had no significant association with variation in family size.

Genotypes at these two candidate sites influenced fertility in opposite directions and were associated with different components of fertility. For the C4BPA candidate site, a larger number of derived male C4BPA 300 Ile alleles was associated with increased fertility (higher birth rate, or shorter interbirth intervals); but for the ZP2 candidate site, a smaller number of derived female ZP2 36 Gly alleles was associated with increased fertility (larger family size, or more births). Although these two candidate sites did not appear to have interactive effects on either family size or birth rate, linkage disequilibrium between them would be consistent with the expectation that selection acts on the functional interaction between C4BPA and ZP2 at fertilization. Using the same measure of genotype associations (the contingency table method) used by Rohlfs, Swanson & Weir (2010), we found significant LD between those two candidate sites in this large sample of Hutterite families (χ2 = 13.72, P = 0.0082). This finding reflected both the low observed frequency of double homozygotes for the two ancestral alleles and the high observed frequency of double homozygotes for the derived C4BPA 300 allele (Ile/Ile) with the ancestral ZP2 36 allele (Val/Val) (Table 5). Overall, the evidence for LD between these two candidate sites and their individual effects on reproductive success are consistent with both our phylogenetic and population genetic analyses, and with the hypothesis that these genes evolve under some form of balancing selection that favours divergent alleles at specific functionally important sites under selection at fertilization.

Table 5 Genotype associations between two candidate sites under selection in a founder population.

Each entry shows the observed (and expected) two-locus genotype frequencies in Hutterite families. Expected frequencies were calculated from genotype probabilities using the genotype association method in Rohlfs, Swanson & Weir (2010).

ZP2 36	C4BPA 300	
	Thr/Thr	Thr/Ile	Ile/Ile	Total	
Val/Val	9 (20.5)	41 (35.5)	20 (13.9)	70	
Val/Gly	151 (138.7)	242 (240.0)	80 (94.2)	473	
Gly/Gly	255 (255.7)	435 (442.4)	182 (173.8)	872	
Total	415	718	282	1,415	

Discussion

Selection on human gamete traits

Genes underlying morphological, behavioural, physiological, and molecular traits that affect human mate choice and reproductive success are expected to evolve in response to sexual selection or sexual conflict over the specificity or choosiness of mating (Andersson & Simmons, 2006; Dixson, 2009), and those traits could include molecules expressed on the surfaces of sperm and eggs (Swanson & Vacquier, 2002; Palumbi, 2009). Behavioural and morphological traits involving premating interactions are widely viewed (e.g., Puts, 2010) as the more important targets of sexual selection on human mate choice compared to molecular traits involving interactions between gametes after mating. However, molecular evidence for selection on genes encoding human sperm-egg binding molecules and other reproductive proteins (Swanson et al., 2001; Swanson & Vacquier, 2002; Hamm et al., 2007; Meslin et al., 2012; Good et al., 2013) indicates that human populations also respond to selection on reproductive success at the level of biochemical and cellular interactions within the female reproductive tract at the time of fertilization (Vacquier, Swanson & Lee, 1997; Swanson & Vacquier, 2002).

Genetic and phenotypic evidence for coevolution under selection

An important area of focus for studies of the human molecular mating system has been zona pellucida glycoproteins on the egg and ZP-binding proteins in sperm (Litscher, Williams & Wassarman, 2009; Wassarman, 2009; Avella, Xiong & Dean, 2013). Using data and methods that complement the population genetic analyses of this system by Rohlfs, Swanson & Weir (2010), we found patterns that are consistent with (1) their hypothesis that LD between two physically unlinked gamete-recognition genes is caused by balancing selection favouring alternative pairs of alleles, and (2) the functional interpretation that such selection may arise from allelic differences at each of those interacting genes through their influence on gamete binding and fertilization success.

The significant functional correlations between maternal or paternal genotypes at candidate sites and variation in both family size and birth rate (Fig. 3) suggest that these candidate sites are indeed targets of selection and are not false positives. In the available data, allelic effects at these two candidate sites in C4BPA and ZP2 appear to be additive, similar to the additive effects of SNP markers on fertility identified through genetic association analyses of Hutterite families (Kosova et al., 2012). Moreover, one of those candidate sites (C4BPA 300) was also discovered as a tag SNP associated with ZP3 variation using a different method (LD calculations) in a different study population (Rohlfs, Swanson & Weir, 2010). The overall consistency in the identification of sites under selection, their associations with reproductive variation from three different methodological approaches (genotypic association, models of codon evolution, LD), the discovery of the same candidate sites using two different codon-model methods (MEME, omegaMap), the specific location of those candidate sites in each gene (especially the single candidate site in the known sperm-binding domain of each ZP gene; Fig. 1), and the distinctive patterns of population genetic variation including strongly balanced polymorphisms (Table 4) that were similar across all three genes (Fig. 2), all point toward selection acting on functional variation involving these three genes and their epistatic interactions.

These interpretations have several corollaries that could form the basis for further tests of the hypothesis of coevolution among candidate sites in these three gamete-recognition genes under selection. The similar patterns of population genetic variation across the three genes suggest that both ZP3 and ZP2 coevolve with C4BPA, but the available data do not allow us to test the functional association between fertility and variation at the ZP3 candidate site. Genotype and fertility data for all three candidate sites including ZP3 315 from large population samples are needed to test that expected association, and compare it to the effects of ZP2 36 variation on fertility. Similarly, LD is expected to be strong between C4BPA 300 and both of the candidate sites in ZP genes, but the available data do not include all three candidate sites genotyped in the same large population samples.

The evidence for functional and evolutionary covariation between ZP genes and C4BPA is also consistent with evidence that those ZP gene products interact as a complex in the zona pellucida. Purified ZP3 in solution forms a homodimer complex in which a pair of proteins binds in antiparallel orientation (Han et al., 2010), and this head-to-tail dimerization brings the N-terminal region of one protein into contact with the C-terminal region of the other (including the C-terminal sperm-binding domain of ZP3 that shows evidence of selection). Moreover, ZP3 expressed in the zona pellucida is hypothesized to form a heterodimer with ZP2 via the same antiparallel orientation of their N- and C-terminal regions (Darie et al., 2008; Monné & Jovine, 2011; Clark, 2013; Wassarman & Litscher, 2013). Such ZP3-ZP2 heterodimers could form a single sperm-binding structure that includes both ZP3 315 and ZP2 36, interacts with ZP-binding proteins on the sperm head (including C4BPA), and mediates coevolution among all three genes via compensatory changes (e.g., Hughes, 2012) between ZP3 and ZP2. That functional interpretation is supported by recent observations that inherited mutations in human ZP3 and ZP2 cause infertility in double heterozygotes, and that mouse models of those double mutants are infertile in part due to susceptibility to polyspermy in laboratory fertilizations (Liu et al., 2017). Additional structural information for ZP proteins and their polymeric organization (e.g., Han et al., 2010) could form the basis for testing the predicted functional significance of the three candidate sites under selection that were identified here.

Our functional interpretation of the genetic evidence for selection on these three human genes is based mainly on mouse studies. An important assumption of that interpretation is that the mouse and human genes are orthologous. There is little doubt that mouse Zp3 and human ZP3 (and mouse Zp2 and human ZP2) are orthologs, but orthology between mouse Zp3r and human C4BPA is unlikely. The two genes have similar coding sequence organization (including a series of 7 or 8 sushi domains), and occur in similar locations on chromosome 1 relative to other genes in the RCA gene cluster; in the mouse genome, Zp3r is adjacent to a C4 binding protein gene (called C4bp in mouse, and C4BPA in humans), and to a C4bp-like pseudogene (C4bp-ps1). However, in the rat genome this part of the RCA gene cluster includes three functional genes: Zp3r, C4bpa, and C4bpb. These comparisons to the rat genome suggest that mouse C4bp is orthologous with rat C4bpa and human C4BPA; C4bpb has become a pseudogene in the mouse (C4bp-ps1); and that the rodent gene expressed in the sperm acrosomal vesicle (Zp3r) is not found in humans and may be descended from a gene-duplication event in a common ancestor shared by mice, rats, and other rodents but not shared with humans, other primates, or other mammalian lineages.

The possibility that mouse Zp3r is not orthologous with human C4BPA adds a complication to our interpretation of the genetic evidence for selection on both sperm- and egg-expressed genes in humans. A single positively-selected codon in C4BPA (C4BPA 300, rs2075520) is in linkage disequilibrium with SNPs in both ZP3 (Rohlfs, Swanson & Weir, 2010) and ZP2 (Table 5); shows population genetic evidence of selection (Table 3); and shows genealogical patterns of haplotype variation similar to both ZP3 and ZP2 (Fig. 2). We propose that those patterns are the consequence of selection acting on functional interactions between human C4BPA and ZP proteins at fertilization, similar to previously documented interactions between ZP3R in the mouse sperm acrosome and ZP proteins in the mouse zona pellucida. That proposal, and our analyses of human C4BPA evolution (and coevolution with ZP3 and ZP2), is not dependent on orthology between C4BPA and Zp3r. However, functional studies of expression of C4BPA in human sperm, and of interactions between human C4BPA and ZP proteins in the human egg coat at fertilization, are needed in order to more fully understand the roles of those genes in both primate and rodent fertilization.

Significance of coevolution among human fertilization genes

Molecular evidence for selection on balanced polymorphisms in human gamete-binding genes is potentially important because it implicates sexual selection in the evolution of human fertility and identifies possible causes of clinical infertility (e.g., Männikkö et al., 2005; Pökkylä et al., 2011). However, other possible sources of selection should also be considered. Expression of C4BPA is well known to be involved in regulation of the classical complement pathway in human immunity (Müller-Eberhard & Miescher, 1985; Krushkal, Kemper & Gigli, 1998), and is implicated in immune responses to viral pathogens including HIV (Chang et al., 2012). Numerous codons were identified as targets of positive selection in divergence of C4BPA among human and other primate species (Cagliani et al., 2016). Balanced polymorphisms may be common among human genes involved in host-pathogen interactions (Leffler et al., 2013; DeGiorgio, Lohmueller & Nielsen, 2014; Azevedo et al., 2015; Cagan et al., 2016). In this context, environmental variation in selection by pathogens could favour multiple C4BPA alleles (e.g., in African and non-African populations) when C4BPA alleles are expressed in the complement pathway, while the same C4BPA polymorphisms could lead to balanced polymorphisms in ZP3 and ZP2 caused by selection favouring egg-sperm allele or genotype matching when C4BPA alleles are expressed in sperm. That hypothesis is consistent with recent evidence for genome-wide positive selection on primate genes (including C4BPA) that encode proteins involved in innate immunity (Van der Lee et al., 2017). In that study, positive selection for nonsynonymous substitutions among primate species was correlated with the accumulation of nonsynonymous polymorphisms within the human lineage in the same genes (including polymorphisms in C4BPA), and was interpreted as evidence of ongoing selection among humans driven by viruses and other pathogens. Moreover, among the three candidate sites analyzed in our study, the largest allele-frequency difference among population groups in the 1000 Genomes Phase 1 data was found at C4BPA 300, which has a four-fold difference in the minor allele frequency (Table 4). That large difference is consistent with the possibility that the evolution of this three-gene system may be driven in part by pleiotropic selection on other effects of C4BPA polymorphisms such as in innate immunity. How this pleiotropy might interact with selection on the effects of C4BPA in fertility and its coevolution with ZP genes is not known.

Candidate sites under selection may be ancient balanced polymorphisms

The hypothesis of coevolution based on balancing selection and allelic matching between male- and female-expressed gamete-recognition genes (Rohlfs, Swanson & Weir, 2010) is also consistent with limited data from the genomes of archaic humans in Eurasia, in which the occurrence of these same polymorphisms at candidate sites under selection suggests that the polymorphisms may be older than the origins of early modern humans in Africa. Among the relatively low-coverage genomes for several Neanderthal individuals (Green et al., 2010) in the UCSC genome browser, no sequence reads are mapped to the candidate sites in C4BPA (rs4844573) or ZP3 (rs2906999); one of those genomes (Vi33.16) includes several sequence reads that map to the candidate site in ZP2 (rs2075520), and all of those reads include the derived allele (ZP2 36 Gly). However, in the more recent high-coverage genome for a Neanderthal individual (Prüfer et al., 2014), there is a mix of ancestral and derived alleles at these sites: two ancestral C4BPA 300 alleles (a Thr-Thr homozygote); one ancestral and one derived ZP3 315 allele (a Pro-Ser heterozygote); and two derived ZP2 36 alleles (a Gly-Gly homozygote). Similarly, in the high-coverage genome for a Denisovan individual (Meyer et al., 2012), multiple reads map to all three of those candidate sites, and all reads encode the derived allele for each site. The occurrence of both ancestral and derived alleles in archaic humans outside of Africa is similar to variation at the same candidate sites in modern human genomes, and is consistent with the hypothesis that these balanced polymorphisms have been maintained for an evolutionarily long time, perhaps as long as the early-Pleistocene divergence between archaic and modern humans (>500,000 years; see Kelso & Prüfer, 2014). That pattern of shared alleles between archaic and modern humans is similar to the pattern of shared Zp3 and Zp2 haplotypes between closely related mouse species (Turner & Hoekstra, 2006) that were interpreted as the products of balancing selection acting on long time scales and leading to trans-specific polymorphisms (Turner & Hoekstra, 2008a; Turner & Hoekstra, 2008b). That pattern is also similar to (but not as old as) trans-specific polymorphisms shared between extant humans and other extant great apes that have been attributed to the effects of balancing selection (Azevedo et al., 2015). The parallel patterns of balanced polymorphisms at all three genes studied here indicate possibly important effects of selection on the expression and function of those gene products during human fertilization.

Supplemental Information

Supplemental Information 1 Workflow and code used to download and analyze haplotypes from the 1000 Genomes Project phase 1

The file includes a narrative description of the work flow, and scripts used to download and format haplotypes for each gene using SNP data in VCF format downloaded from the 1000 Genomes database.

Click here for additional data file.

Table S1 Haplotypes of variable amino acids for three genes

A table of all unique haplotypes is given for each gene. Numbers above each table indicate variable codons numbered from the start codon (1) in each gene. Entries show single-letter amino acids at each variable site for each of the unique haplotypes in the 1000 Genomes phase 1.

Click here for additional data file.

Supplemental Information 3 Population summary statistics for birth rate and family size in a founder population

Birth rates (per day) are calculated from average interbirth interval within each family for all families with 2 or more children (341 total). Family sizes are calculated for all families (361 total). These are the data used in the GEMMA analysis to estimate residual birth rates or residual family sizes associated with polymorphisms at candidate sites under selection (Fig. 3).

Click here for additional data file.

Supplemental Information 4 McDonald-Kreitman tests of balancing selection in three human gamete-recognition genes

Each table shows the number of synonymous or nonsynonymous polymorphic sites within humans (in the 1000 Genomes Project phase 1), and the number of synonymous or nonsynonymous fixed differences between humans and a single chimpanzee haplotype (accession numbers for chimpanzee genes are given in the main text). The chi-squared value for each two-by-two contingency table, and the associated P-value, are shown for each gene.

Click here for additional data file.

Thanks to M Elliot (University of Cambridge) for scripts and help in formatting data from the 1000 Genomes Project; and to F Breden and B Crespi (Simon Fraser University) for advice and comments.

Additional Information and Declarations

Competing Interests

Author Contributions

Data Availability

The authors declare there are no competing interests.

Michael W. Hart conceived and designed the experiments, analyzed the data, wrote the paper, prepared figures and/or tables, reviewed drafts of the paper.

Daryn A. Stover, Vanessa Guerra, Sahar V. Mozaffari and Carina F. Mugal analyzed the data, wrote the paper, prepared figures and/or tables, reviewed drafts of the paper.

Carole Ober and Ingemar Kaj wrote the paper, reviewed drafts of the paper.

The following information was supplied regarding data availability:

A text file that summarizes the workflow and code that was used to download and format haplotype data from the 1000 Genomes Project Phase 1 database, and a text file that contains summary statistics on fertility in Hutterite families, have been provided as Supplemental Files. These are population-level summaries only.

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
