# Peer review of "Positive selection on human gamete-recognition genes"

_PeerJ, doi:10.7717/peerj.4259_

## Round 0.1 · original submission · Minor Revisions

My apologies for the delay in returning a decision on your manuscript - we had to replace a reviewer - but based on my own reading and the comments from the reviewers, I am pleased to suggest only minor revisions to your manuscript. Reviewer #3 in particular had some good insights and suggestions for additional numbers that could be reported and clarifications.

I found it a very interesting and well-written paper overall, and I hope that you consider resubmitting a revised version of this manuscript.

Reviewer 1 ·

Basic reporting

.

Experimental design

.

Validity of the findings

.

Additional comments

I have read the paper by Hart et al. The authors have tried to identify specific sites under selection that could account for previously discovered linkage disequilibrium (LD) between human C4BPA and ZP3 and extend the analyses to include human ZP2. I find that the paper is interesting and it is generally well-written. However, I have several comments which can help improve the paper.

Why rs2286428 and rs200481427 are no in Table 1?

Page 14, line 293, middle, Table 2 should be Table 1.

Page 15, bottom paragraph, the authors list the test methods for selection, but only Vitti, Grossman & Sebati 2013 is referred. To me, the references of Tajima’s D, the composite likelihood ratio test (CLR), and Fay and Wu’s H, as well as xp-EHH should be given too. Although Vitti, Grossman & Sebati 2013 was published in 2013, it missed the latest method EHHST and xp-EHHST developed in

1. Zhong M, Zhang YW, Lange K, and Fan RZ (2011) A cross-population extended haplotypebased
homozygosity score test to detect positive selection in genome-wide scans. Statistics and
Its Interface 4:51-63.

2. Zhong M, Lange K, Papp JC, and Fan RZ (2010) Extended homozygosity score tests to detect
positive selection in genome-wide scans. European Journal of Human Genetics 18:1148-1159.

EHHST and xp-EHHST has a better feature of easy to calculate and easy to interpret results than xpEHH. Hence, the authors should add the results of these better methods to Table 3. The related software can be found on https://sites.google.com/a/georgetown.edu/ruzong-fan/about

The plots of Figure 3 are not clear to me. I wonder if the mean or median allele numbers can be provided to make it easy to understand.

In Table 3, the p-values should be provided in addition to the test statistic values.

Minor issues: in references, there are some problems. For example, the tauthors of following ref

Wang H, Choi Y, Bamidele T, Xuefeng W, Morris N, Zhang X, Broeckel U, Hanis C,
Iardia S, Redline S, Cooper RS, Tang H, Zhu X. 2017. Genome-wide survey in African
Americans demonstrates potential epistasis of fitness in the human genome. Genetic
Epidemiology 958 41:122–135.

should be:

Wang H, Choi Y, Tayo B, Wang X, Morris N, Zhang X, Broeckel U, Hanis C, Kardia S,

Reviewer 2 ·

Basic reporting

Overall, this report reads well with professional English, well thought out hypothesis and supporting evidence for them. Although I am not entirely familiar with the field, the included references appear to be correct and overall attribution seems appropriate.

My only suggestion is to include significance values in Fig. 3 and to more clearly delineate their findings from previous reports, especially in the discussion.

Experimental design

No comment

Validity of the findings

No comment

·

Basic reporting

Overall well written and referenced, see general comments for some specific suggestions for improvement.

It would be helpful to provide a link in the fertility data supplementary file to the location of the previously published values for individual families, on which the summary stats are based.

Experimental design

Questions are clearly defined and methods appropriate. See suggestions for some additional statistical tests in general comments.

I particularly appreciate the acknowledgment of 'researcher degrees of freedom' and possibility for inflated false positives. The authors do a nice job clearly showing the 'story' of their study.

Validity of the findings

The authors are clear about the relative robustness of their findings and which should be treated with caution.

Additional comments

Hart and colleagues investigate evolution of egg and sperm binding proteins in humans. They test for evidence of positive selection on two egg coat proteins ZP2 and ZP3, consistent with findings reported in other mammals. Furthermore, they test for positive selection on C4BPA the sperm protein which is the putative human homolog of mouse Zp3r, which binds to the egg coat. They find evidence for positive selection on one or more amino acid sites in each of these proteins, but not in the egg coat protein ZP1. Moreover, they report associations between the candidate selected sites in ZP2 and C4BPA with fertility in a Hutterite population.

This is an interesting study testing implications of the pattern of rapid adaptive evolution of fertilization proteins in mammals (and other taxa) - usually viewed in the context of reproductive isolation and speciation - for reproductive compatibility within humans. The authors do a nice job explaining the limitations of the methods employed for within species data, and are clear that their findings are suggestive and should be treated with caution - but are exciting leads for future research.

1. I found the manuscript well written and easy to follow, with the exception of the abstract - the framing of the question was somewhat muddled. I suggest rewriting the beginning of the abstract and streamlining the description of results. For example the intro focuses on co-evolution between egg and sperm proteins, but that's not clear from the abstract - which starts focused on egg proteins.

2. population genetics analysis
A. it is unclear to me why it was not possible to perform analyses on the ZP3 site using downloaded variable sites. Most/all of the test statistics can by computed using other programs or calculated directly. Was the author/administrator of the 1000 genomes selection browser contacted to correct the error in order to run analysis?
Please include analysis for ZP3 using other software (at least Fst, TajD, Fay & Wu's H, CLR) or explain in more detail why this wasn't possible.
B. I suggest adding McDonald-Kreitman (Nature 1991) tests. While conservative, this test is commonly used to distinguish high amino acid polymorphism due to balancing selection vs. relaxed purifying selection. As other great ape sequences were used to identify ancestral vs. derived mutations, appropriate outgroup sequences are available for MK. πR and πS (replacement and silent diversity) values should be reported for each gene.
C. It would be useful to add popgen tests for the combined sample, in addition to individual populations, to evaluate the hypothesis that divergent alleles are maintained over time in humans. (although I agree with argument from L379 that this may be problematic for LD-based tests)
D. Fst is not a test for selection - change in description of tests and results (beginning L348) to indicate evidence is only for population differentiation not selection

3. positive vs. diversifying vs. balancing selection - Unfortunately, this terminology is not used consistently among different people in the field and some clarification is needed here. It seems in the MEME paper, "diversifying" and "positive" selection are used interchangeably (and for interspecific comparisons). Diversifying sensu Murrell et al 2012 is not the same as balancing selection, but it seems in this manuscript the authors suggest that significant results in MEME are evidence for balancing selection. Be careful to distinguish exactly what the test shows - i.e. rapid & adaptive amino acid substitution at a site - particularly for this intraspecific analysis.
- Title, L13, L16 - change 'selection' to 'positive selection'

Add haplotype tables including variable amino acid sites for each gene as supplementary.

Table 2 - add #s of nonsynonymous and syn polymorphisms for each gene.

Minor comments:
L16 - there have been many papers on sperm proteins too - e.g. reviewed in Turner & Hoekstra IJDB 2008

L18 - change to "selection the human egg-coat genes ZP3 and ZP2. We also found evidence for positive selection on C4BPA, which encodes a repetitive..."

L22 "Several additional lines..." this sentence is confusing right after saying no selection on ZP1 - re-word

L55 - is this the right reference? title still calls it sp56

L140 "one candidate site under diversifying selection" to "one candidate site each under"

L178 - caveats about applying these models within species are explained clearly later in the manuscript. It would be helpful to either move them here or add "(but see caveats below)" or similar.

L706 could add mention of parallel amino acid changes at same sites in ZP3 in multiple rodent taxa and mammals, reviewed in Turner & Hoekstra IJDB 2008

Fig 1 - add x axis with aa positions

Fig 2 - add colors to indicate % alleles from different populations

Fig 3 - smaller, jittered points; add P vals to graph or legend

---

## Round 0.2 · accepted · Accept

Happy holidays! We're delighted to let you know that your article has been accepted for publication. Thank you for your efforts in revising, and addressing the reviewer comments.

Your change to the author list to include Vanessa Guerra has been noted, and will be reflected in the final version of the manuscript as you have indicated.

·

Basic reporting

good

Experimental design

good

Validity of the findings

good

Additional comments

I am satisfied by the changes made in response to my comments and those of other reviewers, or the clear and thorough explanations why they were not possible/advisable. I appreciate in particular the extra work necessary to run selscan to run haplotype tests for ZP3.